 

# The auditory representation of speech sounds in human motor cortex

Connie Cheung[1,2,3,4†], Liberty S Hamilton[2,3,4†], Keith Johnson[5], Edward F Chang[1,2,3,4*]

[1]Graduate Program in Bioengineering, University of California, Berkeley-University of California, San Francisco, San Francisco, United States; [2]Department of Neurological Surgery, University of California, San Francisco, San Francisco, United States; [3]Center for Integrative Neuroscience, University of California, San Francisco, San Francisco, United States; [4]Department of Physiology, University of California, San Francisco, San Francisco, United States; [5]Department of Linguistics, University of California, Berkeley, Berkeley, United States

**Abstract** In humans, listening to speech evokes neural responses in the motor cortex. This has been controversially interpreted as evidence that speech sounds are processed as articulatory gestures. However, it is unclear what information is actually encoded by such neural activity. We used high-density direct human cortical recordings while participants spoke and listened to speech sounds. Motor cortex neural patterns during listening were substantially different than during articulation of the same sounds. During listening, we observed neural activity in the superior and inferior regions of ventral motor cortex. During speaking, responses were distributed throughout somatotopic representations of speech articulators in motor cortex. The structure of responses in motor cortex during listening was organized along acoustic features similar to auditory cortex, rather than along articulatory features as during speaking. Motor cortex does not contain articulatory representations of perceived actions in speech, but rather, represents auditory vocal information.

*For correspondence: edward.chang@ucsf.edu

†These authors contributed equally to this work

Competing interests: The authors declare that no competing interests exist.

## Introduction

Our motor and sensory cortices are traditionally thought to be functionally separate systems. However, an accumulating number of studies has revealed their roles in action and perception to be highly integrated (*Pulvermüller and Fadiga, 2010*). For example, a number of studies have demonstrated that both sensory and motor cortices are engaged during perception (*Gallese et al., 1996*; *Wilson et al., 2004*; *Tkach et al., 2007*; *Cogan et al., 2014*). In humans, this phenomenon has been observed in the context of speech, where listening to speech sounds evokes robust neural activity in the motor cortex (*Wilson et al., 2004*; *Pulvermüller et al., 2006*; *Edwards et al., 2010*; *Cogan et al., 2014*). This observation has re-ignited an intense scientific debate over the role of the motor system in speech perception over the past decade (*Lotto et al., 2009*; *Scott et al., 2009*; *Pulvermüller and Fadiga, 2010*).

One interpretation of the observed motor activity during speech perception is that "the objects of speech perception are the intended phonetic gestures of the speaker"- as posited by Liberman's motor theory of speech perception (*Liberman et al., 1967*; *Liberman and Mattingly, 1985*). The motor theory is a venerable and well-differentiated exemplar of a set of speech perception theories that we could call 'production-referencing' theories. Unlike motor theory, more modern production referencing theories do not assume that sensorimotor circuits are necessarily referenced in order for speech to be recognized, but they allow for motor involvement in perception in certain phonetic

**eLife digest** When we speak, we force air out of our lungs so that it passes over the vocal cords and causes them to vibrate. Movements of the jaw, lips and tongue can then shape the resulting sound wave into speech sounds. The brain's outer layer, which is called the cortex, controls this process. More precisely, neighboring areas in the so-called motor cortex trigger the movements in a specific order to produce different sounds.

Brain imaging experiments have also shown that the motor cortex is active when we listen to speech, as well as when we produce it. One theory is that when we hear a sound, such as the consonant 'b', the sound activates the same areas of motor cortex as those involved in producing that sound. This could help us to recognize and understand the sounds we hear.

To test this theory, Cheung, Hamilton et al. studied how speech sounds activate the motor cortex by recording electrical signals directly from the brain's surface in nine human volunteers who were undergoing a clinical evaluation for epilepsy surgery. This revealed that speaking activates many different areas of motor cortex. However, listening to the same sounds activates only a small subset of these areas. Contrary to what was thought, brain activity patterns in motor cortex during listening do not match those during speaking. Instead, they depend on the properties of the sounds. Thus, sounds that have similar acoustic properties but which require different movements to produce them, such as 'b' and 'd', activate the motor cortex in similar ways during listening, but not during speaking.

Further research is now needed to work out why the motor cortex behaves differently when we hear as opposed to when we speak. Previous work has suggested that the region increases its activity during listening when the sounds heard are unclear, for example because of background noise. One testable idea therefore is that the motor cortex helps to enhance the processing of degraded sounds.

modes. For example, *Lindblom, 1996* suggested that a direct link between spectrotemporal analysis and word recognition is the normal mode of speech perception (the 'what' mode of perception), but in some cases listeners do use a route through sensorimotor circuits (the 'how' mode of perception) if, for example, the listener is attempting to imitate a new sound (*Lindblom, 1996*).

While demonstrations of evoked motor cortex activity by speech sounds strengthen production-referencing theories, it remains unclear what information is actually *represented* by such activity. Determining what phonetic properties are encoded in the motor cortex has significant implications for elucidating the role it may play in speech perception. To address this, we recorded direct neural activity from the peri-Sylvian speech cortex in nine human participants undergoing clinical monitoring for epilepsy surgery. This includes but is not limited to two relevant areas comprising the supra-Sylvian ventral half of the lateral sensorimotor cortex (vSMC) for the motor control of articulation (*Penfield and Boldrey, 1937*) and the infra-Sylvian superior temporal gyrus (STG) for the auditory processing of speech sounds (*Ojemann et al., 1989*; *Boatman et al., 1995*). Since cortical processing of speech sounds is spatially discrete and temporally fast (*Formisano et al., 2008*; *Chang et al., 2011*; *Steinschneider et al., 2011*), we used customized high-density electrode grids (a four-fold increase over conventional recordings) (*Bouchard et al., 2013*; *Mesgarani et al., 2014*). Importantly, these recordings have simultaneous high spatial and temporal resolution in order to study the detailed speech representations in the vSMC (*Crone et al., 1998*; *Edwards et al., 2009*). With this approach, we seek to address unanswered questions about the representation of speech sounds in motor cortex, including how the spatiotemporal patterns compare when speaking and listening and whether auditory representations in motor cortex are organized along articulatory or acoustic dimensions.

## Results

Participants first listened passively to consonant-vowel (CV) syllables (8 consonants followed by the /a/ vowel). In a separate trial block, they spoke aloud these same CV syllables. We measured the average evoked cortical activity during these listening and speaking CV tasks. We focused our

analysis on high gamma (70–150 Hz) cortical surface local field potentials, which strongly correlate with extracellular multi-unit neuronal spiking (*Steinschneider et al., 2008*; *Ray and Maunsell, 2011*). We aligned neural responses to the onset of speech acoustics (t = 0) in listening and speaking tasks to provide a common reference point across speech sounds.

We first determined which peri-Sylvian cortical areas were activated during passive listening to speech sounds. *Figure 1a and b* shows the locations of cortical areas that demonstrated cortical evoked responses in a single representative subject during listening and speaking respectively. During listening, evoked responses spanned middle and posterior STG as expected, with weaker responses in middle temporal gyrus (MTG) (*Figure 1a*). In the vSMC, (composed of the pre- and post- central gyri) we found electrodes in the superior-most and inferior-most aspects (*Figure 1a*, *Figure 1—figure supplement 1*, *2*) that demonstrated reliable and robust single-trial responses to speech sounds during passive listening (*Figure 1b*). Neural responses were also found at a few sites scattered across supramarginal, inferior-, and middle- frontal gyri—though these were not consistent across subjects (*Figure 1—figure supplement 1*). By performing spatial clustering analysis on the electrode positions in each subject, we found that 3/5 subjects showed significant clustering of regions responsive to auditory stimuli (Hartigan's Dip statistic, p<0.05 (see Materials and methods); *Figure 1—figure supplement 1*). Out of these 3 subjects, k-means clustering revealed two subjects with k=2 electrode clusters (subjects 1 and 4, clusters in inferior and superior vSMC), and one subject with k=5 clusters. When participants spoke the same CV syllables, in contrast, articulatory movement-related cortical activity was well distributed throughout vSMC (*Figure 1c*), with auditory feedback cortical activity seen in the STG.

Across all participants, we identified 115 electrodes that demonstrated significant neural activity in vSMC during listening (p<0.01, t-test, compared to pre-stimulus silent rest period; *Figure 1d*). When speaking, in contrast, a total of 362 electrodes in vSMC were found to be significantly active (*Figure 1d*, p<0.01, t-test, compared to pre-stimulus silent rest period). We compared the relative proportions of electrodes that were found in different supra-Sylvian anatomical regions. Critically, only a subset of sites in vSMC (98 out of 362, ∼27%) was active during both listening and speaking (*Figure 1d*). These sites were primarily localized to the pre-central gyrus, whereas speaking evoked activity across both pre- and post-central gyri sites. Neural responses in the vSMC during listening were found in the superior (S in *Figure 1d*) pre-central gyrus and inferior, anterior aspect of the sub-central gyrus of the vSMC (I in *Figure 1d*).

We next compared the patterns of cortical activity to specific speech sounds during listening and speaking. During speaking, specific articulator representations have been identified in the somato-topically-organized vSMC (*Bouchard et al., 2013*). For example, the plosive consonants /b/, /d/, and /g/ are produced by the closure of the vocal tract at the lips, front tongue, and back tongue, respectively (*Figure 2a, b*, see *Figure 2—figure supplement 1* for all syllable tokens) (*Ladefoged and Johnson, 2010*). The cortical representations for these articulators are laid out along a superior-to-inferior (medial-to-lateral) sequence in the vSMC (*Penfield and Boldrey, 1937*). We first examined average cortical activity at single electrode sites distributed along the vSMC axis for articulating individual speech sounds. *Figure 2c* shows single electrode activity from a single representative subject (the same from *Figure 1*) for speaking (blue lines) and listening (red lines) for three CV syllables, which have different place of articulation (/ba/, /da/, and /ga/). The exact location of these electrodes on the vSMC is shown in *Figure 2d*. The production of labial consonants (/b/) is associated with activity in lip cortical representations as evidenced by strong responses to the bilabial /ba/ (*Figure 2c*, electrodes 5–6, blue lines). These are located superior to the tongue representations associated with the /d/ and /g/ consonants, as shown previously (*Bouchard et al., 2013*). Those tongue sites were sub-specified by 'coronal' (i.e. anterior-based) tongue position for /d/ (electrodes 8–10, blue lines) superiorly, and 'dorsal' (i.e. posterior-based) tongue position for /g/ inferiorly (electrode 13, blue line). Other sites (electrodes 1–4, 11–12, blue lines) showed the same neural activity across all three syllables.

We next examined those same vSMC electrodes during listening, and found that the majority of those cortical vSMC electrodes were not active (p>0.01, t-test compared to silence, *Figure 2c* transparent red lines). The few that were active (electrodes 1, 2, 4, 11–12, solid red lines) were similar for all three CV syllables, with activity increasing approximately 100ms after the acoustic onset. Across the entire population of vSMC electrodes that were active during listening, onset latencies were generally shorter than those in STG sites, with significant increases in both inferior vSMC (p<0.001)

**Figure 1.** Speech sounds evoke responses in the human motor cortex. (**a**) Magnetic resonance image surface reconstruction of one representative subject's cerebrum (subject 1: S1). Individual electrodes are plotted as dots, and the average cortical response magnitude (z-scored high gamma activity) when listening to CV syllables is signified by the color opacity. CS denotes the central sulcus; SF denotes the Sylvian fissure. (**b**) Acoustic waveform, spectrogram, single-trial cortical activity (raster), and mean cortical activity (high gamma z-score, with standard error) from two vSMC sites and one STG site when a subject is listening to /da/. Time points significantly above a pre-stimulus silence period (p<0.01, bootstrap resampled, FDR corrected, alpha < 0.005) are marked along the horizontal axis. The vertical dashed line indicates the onset of the syllable acoustics (t=0). (**c**) Same subject as in (**a**); distributed vSMC cortical activity when speaking CV syllables (mean high gamma z-score). (**d**) Total number of significantly active sites in all subjects during listening, speaking, and both conditions (p<0.01, t-test, responses compared to silence and speech). Electrode sites are broken down by their anatomical locations. S denotes superior vSMC sites; I denotes inferior vSMC sites.

The following figure supplements are available for figure 1:

**Figure supplement 1.** Average cortical responses to speaking and listening in all subjects (S2-S5).

**Figure supplement 2.** Neural responses while listening to CV syllables in 4 additional subjects not included in MDS analyses (S6 - S9).

and superior vSMC (p<0.05) compared to STG (*Figure 3a*, Wilcoxon rank sum test, see *Figure 3c* for average responses to all syllables). The latency to the response peak was also significantly higher in superior vSMC compared to STG (*Figure 3b*, p<0.01, Wilcoxon rank sum test). A cross-correlation analysis between these vSMC electrodes and STG electrodes revealed a diverse array of relationships between these populations (*Figure 3d–f*), including STG electrode activity leading vSMC

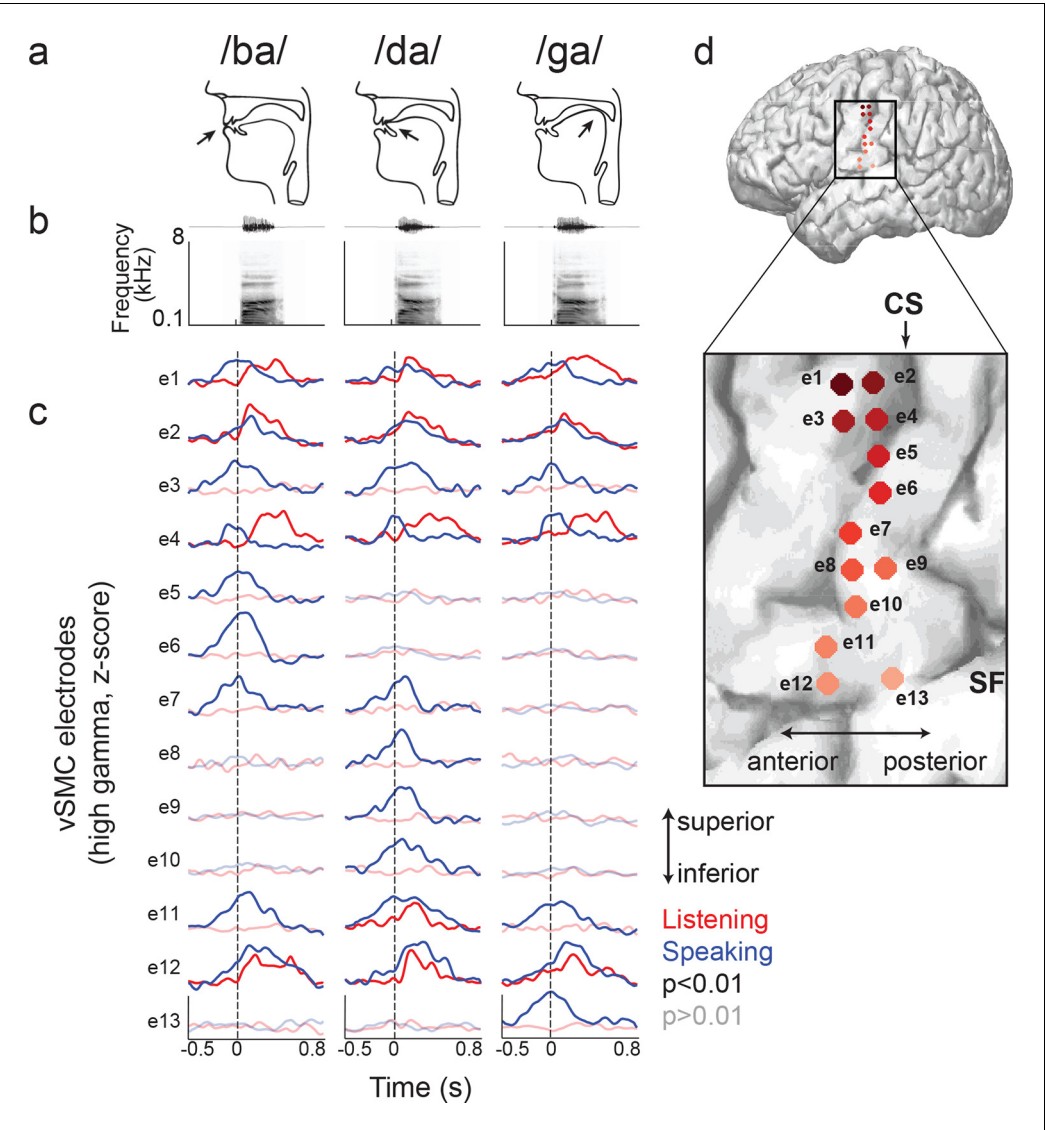

**Figure 2.** Site-by-site differences in vSMC neural activity when speaking and listening to CV syllables. (**a**) Top, vocal tract schematics for three syllables (/ba/, /da/, /ga/) produced by occlusion at the lips, tongue tip, and tongue body, respectively (arrow). (**b**) Acoustic waveforms and spectrograms of spoken syllables. (**c**) Average neural activity at electrodes along the vSMC for speaking (blue) and listening (red) to the three syllables (high gamma z-score). Solid lines indicate activity was significantly different from pre-stimulus silence activity (p<0.01). Transparent lines indicate activity was not different from pre-stimulus silence activity (p>0.01). Vertical dashed line denotes the onset of the syllable acoustics (t=0). (**d**) Location of electrodes 1–13 in panel c, shown on whole brain and with inset detail. CS = central sulcus, SF = Sylvian fissure.

The following figure supplement is available for figure 2:

**Figure supplement 1.** Syllable token set.

electrode activity and vice versa. In contrast to speaking, we did not observe somatotopic organization of cortical responses when listening to speech. Therefore, the pattern of raw evoked responses during listening shows critical differences from those during speaking.

We next evaluated quantitatively whether the structure of distributed vSMC neural activity during listening was more similar to that of vSMC during speaking or the STG during listening. In previous studies, we demonstrated that the structure of evoked responses are primarily organized by



**Figure 3.** Dynamics of responses during CV listening in STG, inferior vSMC, and superior vSMC. (**a**) STG onset latencies were significantly lower than both inferior vSMC (p<0.001, Z = −4.03) and superior vSMC (p<0.05, Z = −2.28). (**b**) STG peak latencies were significantly lower than superior vSMC (p<0.01, Z = −2.93), but not significantly different from peak latencies in inferior vSMC (p>0.1). In (**a**) and (**b**), red bar indicates the median, boxes indicate 25th and 75th percentile, and error bars indicate the range. Response latencies were pooled across all subjects. All p-values in (**a**) and (**b**) are from the Wilcoxon rank sum test. (**c**) Average evoked responses to all syllable tokens across sites in superior vSMC (n=32), inferior vSMC (n=37), and STG. Responses were aligned to the syllable acoustic onset (t=0). A random subset of STG responses (n=52 out of the 273 that were used in the latency analysis in (**a**) and (**b**)) are shown here for ease of viewing. (**d**) Example cross-correlations between three vSMC electrodes and all STG electrodes in one patient, for a maximum lag of ± 0.75 s. More power in the negative lags indicates a faster response in the STG compared to the vSMC electrode, and more power in the positive lags indicates a faster response in vSMC compared to STG. We observe vSMC electrodes that tend to

*Figure 3 continued on next page*

*Figure 3 continued*
respond later than STG (e248, left panel), vSMC electrodes that tend to respond before STG (e136, middle panel), and vSMC electrodes that respond at similar times to some STG electrodes (e169, right panel). (**e**) Average evoked responses during CV listening for all STG electrodes from this patient and the three vSMC electrodes shown in panel (**d**). Responses were aligned to the syllable acoustic onset (t=0), as in panel (**c**). (**f**) Percentage of sites with STG leading, coactive, or vSMC leading as expressed by the asymmetry index (see Materials and methods). Both inferior and superior vSMC show leading and lagging responses compared to STG, as well as populations of coactive pairs.

different feature sensitivities: place of articulation in the vSMC (***Bouchard et al., 2013***), and manner of articulation in the STG (***Mesgarani et al., 2014***). We visualized the similarity of population activity evoked by different consonants using unsupervised multidimensional scaling (MDS), where the 2-dimensional Euclidean distances between stimuli correspond to the similarity of their neural responses. Visual inspection of MDS plots shows that, during speaking, evoked activity in vSMC clustered into place of articulation features (***Figure 4a***): labials (/b/, /p/), alveolars (/s/, /sh/, /t/, /d/), and velars (/g/, /k/) (***Figure 4b***). In contrast, neural responses during listening did not cluster into the same features (***Figure 4c***). To quantify the degree to which the evoked activity clustered into place of articulation features, we used unsupervised K-means clustering to assign the neural responses to clusters (k=3), and the adjusted Rand Index ($RI_{adj}$) (***Rand, 1971***; ***Hubert and Arabie, 1985***) to measure the degree to which the neural clustering agreed with linguistically defined place of articulation consonant clusters. The $RI_{adj}$ quantifies the degree of agreement between two clustering patterns, where $RI_{adj} = 1$ denotes identical clustering patterns and $RI_{adj} = 0$ denotes independent clustering patterns. We found that while evoked activity during speaking clustered by place of articulation features, activity during listening did not (***Figure 4d***; see ***Figure 4—figure supplement 1*** for moving time window analysis). Even when the vSMC electrode subset was restricted to short-latency vSMC electrodes leading STG activity (as evidenced by a positive asymmetry index in ***Figure 3f***), activity during listening did not cluster according to place of articulation features (***Figure 4—figure supplement 2***). Thus, responses in motor areas during speech perception do not show a spatially distributed representation of speech motor articulator features.

Finding no evidence that major articulator features are either locally or spatially distributed in the vSMC in response to speech sounds, we next compared vSMC responses to population responses in the STG. STG has an acoustic sensory representation of speech that best discriminates speech sounds by manner of articulation features with salient acoustic differences (***Mesgarani et al., 2014***). Using multidimensional scaling, STG spatial patterns during listening showed clustering according to three high-order acoustic features (***Figure 4e***): voiced plosives (/b/, /d/, /g/), unvoiced plosives (/p/, /t/, /k/), and fricatives (/s/, /sh/) (***Ladefoged and Johnson, 2010***). This is consistent with the relational organization derived by analysis of structure in the stimulus acoustics (***Figure 4—figure supplement 3a***), and the structure of STG during speaking (***Figure 4—figure supplement 3b***). With the same analyses, we observed that activity in motor cortex clustered into the same three acoustic features (***Figure 4f***, note this panel is identical to ***Figure 4c*** simply re-colored). Unsupervised K-means clustering analysis confirmed that vSMC activity, during listening, organized into these linguistically defined acoustic feature groups, but was significantly weaker than the organization of STG (p<0.001, Wilcoxon rank-sum, ***Figure 4g***). Importantly, however, clustering by acoustic manner features was significantly stronger than clustering by place features in vSMC electrodes during listening (p<0.001, Wilcoxon rank-sum, ***Figure 4h***). This organization suggests that motor cortex activity during speech perception reflects an acoustic sensory representation of speech in the vSMC that mirrors acoustic representations of speech in auditory cortex.

To further define the acoustic selectivity and tuning of vSMC motor electrodes, participants listened to natural, continuous speech samples from a corpus with a range of American English speakers (***Garofolo et al., 1993***). We fit spectrotemporal receptive field (STRF) models for each vSMC electrode using normalized reverse correlation (see Materials and methods), which describes the spectrotemporal properties of speech acoustics that predict the activity of a single site in motor cortex. To compute the STRF, we calculate the correlation between the neural response at an electrode and the stimulus spectrogram at multiple time lags. The result is then normalized by the auto-

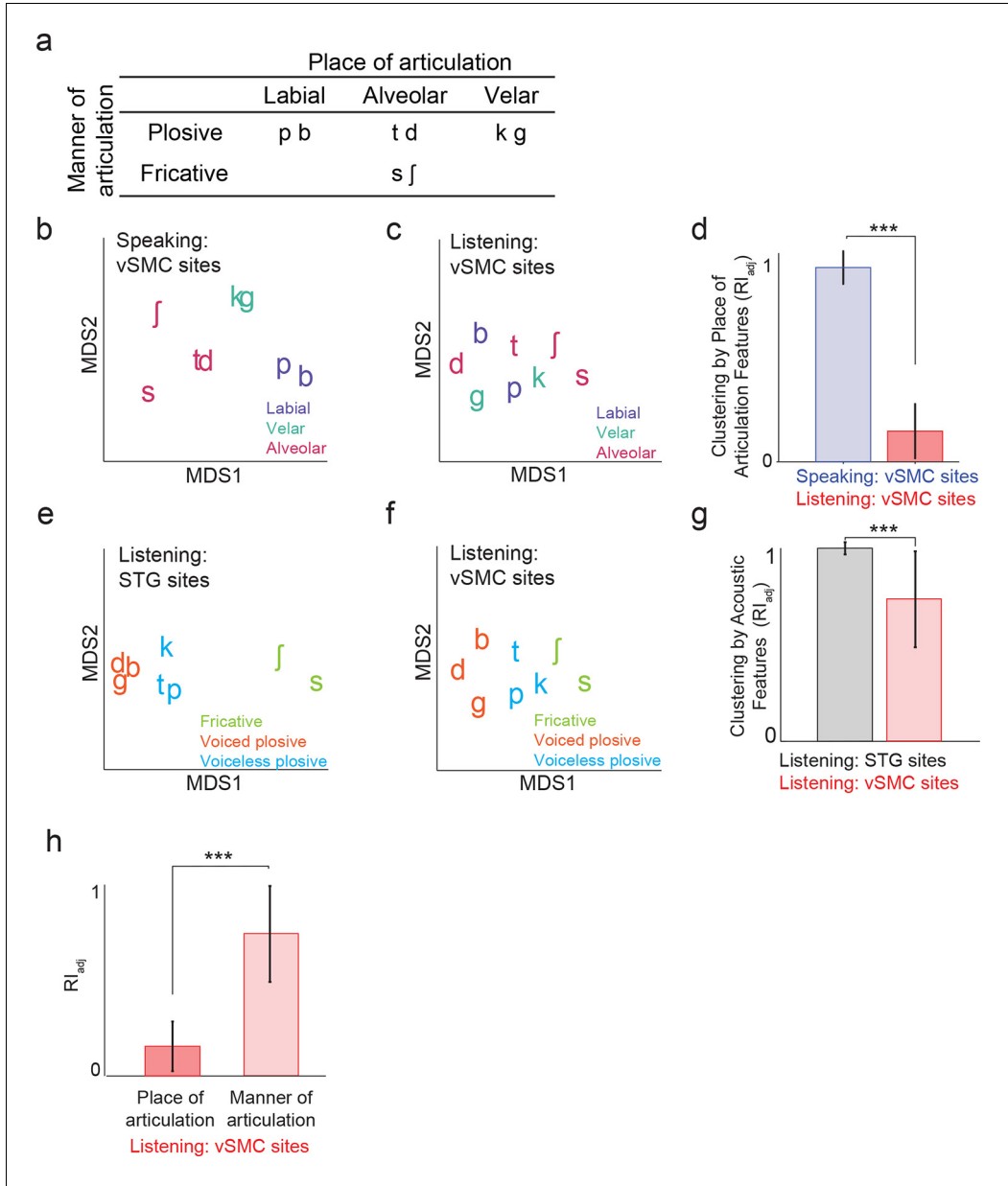

**Figure 4.** Organization of motor cortex activity patterns. (**a**) Consonants of all syllable tokens organized by place and manner of articulation. Where consonants appear in pairs, the right is a voiced consonant, and the left is a voiceless consonant. (**b**) Relational organization of vSMC patterns (similarity) using multidimensional scaling (MDS) during speaking. Neural pattern similarity is proportional to the Euclidean distance (that is, similar response patterns are grouped closely together, whereas dissimilar patterns are positioned far apart). Tokens are colored by the main place of articulation of the consonants (labial, velar, or alveolar). (**c**) Similarity of vSMC response patterns during listening. Same coloring by place of articulation. (**d**) Organization by motor articulators. K-means clustering was used to assign mean neural responses to 3 groups (labial, alveolar, velar) for both listening and speaking neural organizations (**b,c**). The similarity of the grouping to known major articulators was measured by the adjusted Rand Index. An index of 1 indicates neural responses group by place of articulation features. ***p<0.001, Wilcoxon rank-sum (**e**) Organization of mean STG responses using MDS when listening. In contrast to c and d, tokens are now colored by their main acoustic feature (fricative, voiced plosive, or voiceless plosive). (**f**) Organization of mean vSMC responses using MDS when listening colored by their main acoustic feature. (Identical to C, but recolored here by acoustic features). (**g**) Organization by manner of articulation acoustic features (fricative, voiced plosive, voiceless plosive) for both STG and vSMC organizations when listening (**e**, **f**). The similarity of the grouping to known acoustic feature groupings was measured by the adjusted Rand Index. ***p<0.001, Wilcoxon rank sum. (**h**) During listening, responses in vSMC show significantly greater organization by

*Figure 4 continued on next page*

*Figure 4 continued*

acoustic manner features compared to place features as assessed by the adjusted Rand Index, indicating an acoustic rather than articulatory representation (***p<0.001, Wilcoxon rank-sum). Bars in this panel are the same as the red bars in (d) and (g). In (d), (g), and (h), bars indicate mean ± standard deviation,

The following figure supplements are available for figure 4:

**Figure supplement 1.** Clustering trajectory analysis of neural responses to syllables.
**Figure supplement 2.** Analysis of short latency responses in vSMC.
**Figure supplement 3.** Organization of syllable tokens and auditory cortical activity patterns.

---

correlation in the stimulus. This results in a linear filter for each electrode (the STRF), which, when convolved with the stimulus spectrogram, produces a predicted neural response to that stimulus. The prediction performance of each STRF was determined by calculating the correlation between the activity predicted by the STRF and the actual response on held out data. A fraction of vSMC sites (16/98 sites total) were reasonably well-predicted with a linear STRF (r>=0.10 and p<0.01, permutation test) (*Theunissen et al., 2001*). STRFs with significant correlation coefficients were localized to superior and inferior vSMC (primarily precentral gyrus) in addition to STG (*Figure 5a*). Still, the prediction performance of STRFs in vSMC was generally lower than that of the STG (*Figure 5b*). Furthermore, the majority of STRFs in both regions showed strong low frequency tuning (100–200 Hz) properties related to voicing (*Figure 5c*), though some also showed high frequency tuning consistent with selectivity for fricatives and stop consonants by visual inspection (*Mesgarani et al., 2014*). We also estimated the mean cortical response at each motor site to every phoneme in English and found a diverse set of responses (*Figure 5—figure supplement 1a*) that were notably weaker in magnitude compared to STG responses (*Figure 5—figure supplement 1b*). Weak selectivity to phonetic features measured by the Phoneme Selectivity Index (PSI) was also observed (*Figure 5—figure supplement 1c*) (*Mesgarani et al., 2014*). These findings reveal that individual sites in motor cortex reflect sensory responses to definable spectrotemporal features speech acoustics, including voicing attributes. Presumably, this tuning gives rise to the acoustic organization found in the previous analysis of distributed spatial patterns of neural activity.

## Discussion

Our principal objective was to determine the vSMC motor cortex representation of auditory speech sounds. We used high-resolution cortical recordings and a wide array of speech sounds to determine how the vSMC structure of speech sounds compared to the structure of motor commands in vSMC and sensory processing in STG. We found evidence for both spatially local and distributed activity correlated to speech acoustics, which suggests an auditory representation of speech in motor cortex.

The proposal that the motor cortex critically integrates observations with motor commands largely stems from the discovery of mirror neurons (in area F5 of macaques) that fire both when a monkey produced an action and observed a similar action (*di Pellegrino et al., 1992*; *Rizzolatti and Craighero, 2004*; *Pulvermüller and Fadiga, 2010*). This 'integrative' view is reminiscent of linguistic production-referencing theories, including the motor theory of speech perception, which propose that motor circuits are involved in speech perception (*Liberman et al., 1967*; *Liberman and Mattingly, 1985*). In line with these theories, human neuroimaging studies have showed mirror activity in ventral premotor cortex during listening (*Wilson et al., 2004*; *Pulvermüller et al., 2006*; *Edwards et al., 2010*), and modulated premotor activity in phoneme categorization tasks (*Alho et al., 2012*; *Chevillet et al., 2013*). Our results extend these findings by detailing the representational selectivity and encoding of vSMC in perception. Consistent with previous findings, we demonstrated local 'audiomotor' responses to speech sounds in vSMC. When the responses were further examined for phonetic structure, we found major motor articulatory place features, such as labial, alveolar, and velar, were not represented with single site activity or distributed spatial

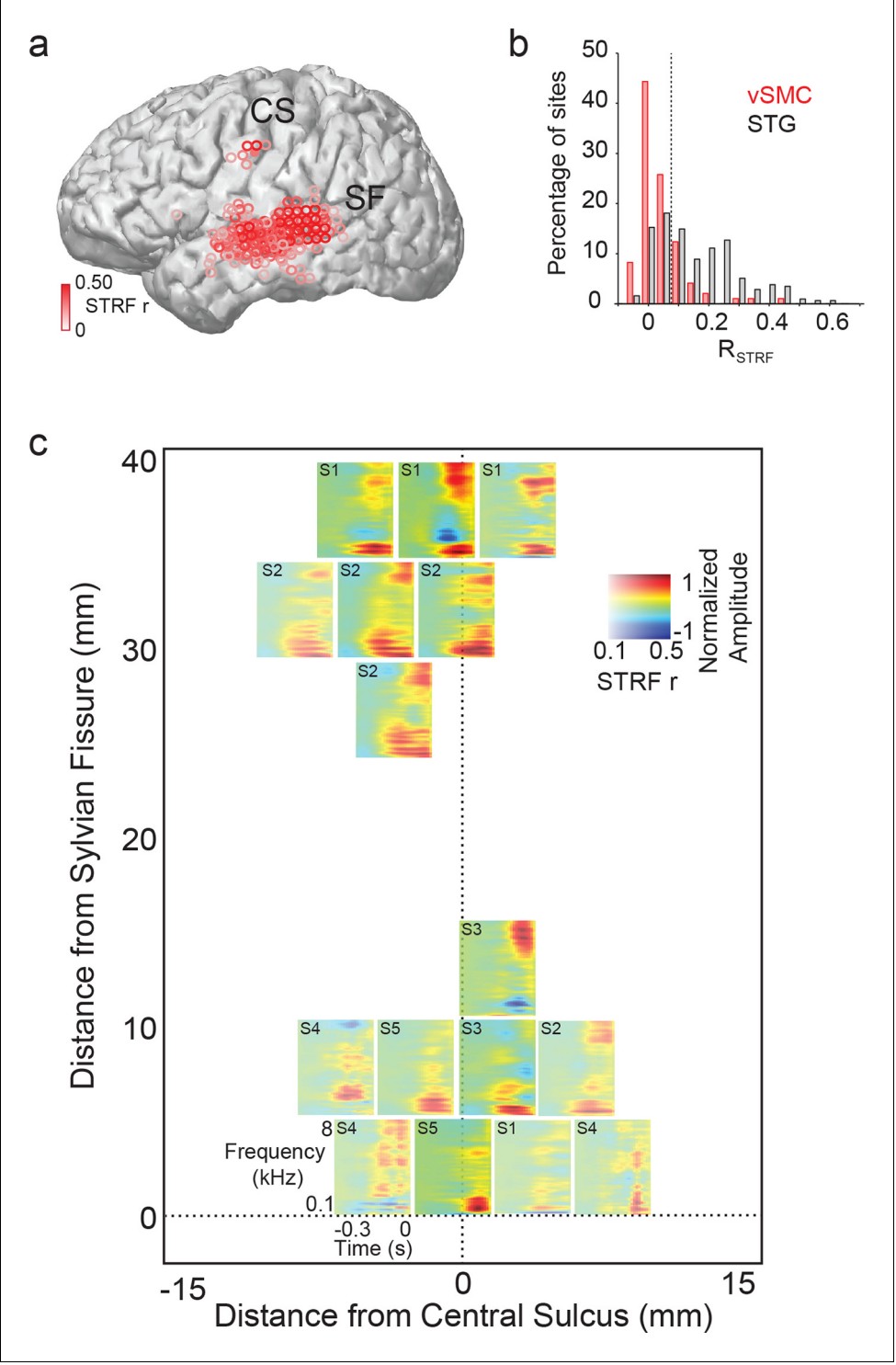

**Figure 5.** Acoustic spectrotemporal tuning in vSMC. (a) All STRF correlations and locations are plotted with opacity signifying the strength of the correlation. CS denotes the central sulcus; SF denotes the Sylvian fissure. (b) Distribution of STRF prediction correlations for significantly active vSMC and STG sites. Cut-off at r = 0.1 is shown as a dashed line. (c) Individual STRFs from all subjects (S1-S5, STRF correlation>0.1) plotted as a function of distance from the central sulcus and Sylvian fissure, with opacity signifying the strength of the STRF correlation.

The following figure supplement is available for figure 5:

**Figure supplement 1.** Summary and comparison of vSMC and STG responses to syllables.

activity. This observation is in direct contrast with structural predictions made by the original motor theory of speech perception (*Liberman et al., 1967*; *Liberman and Mattingly, 1985*), while confirming that motor cortex plays a role in perception (*Lindblom, 1996*; *Hickok and Poeppel, 2007*).

We localized activity during speech perception to regions of the vSMC that have been implicated in phonation and laryngeal control (*Penfield and Boldrey, 1937*; *Brown et al., 2008*). When listening to speech, we observed these regions reflected acoustic sensory properties of speech, with individual sites tuned for spectrotemporal acoustic properties. The tuning properties of responsive sites in vSMC are similar to properties observed in STG during listening (*Mesgarani et al., 2014*) and appear to give rise to an acoustic sensory organization of speech sounds (rather than purely motor organization) in motor cortex during listening.

There is an emerging consensus that frontal and motor regions are recruited during effortful listening (*Du et al., 2014*). For example, previous studies have demonstrated that frontal areas come online to process degraded speech for the attentional enhancement of auditory processing (*Wild et al., 2012*). Our results may complement this interpretation in that the audiomotor cortex enhancement is specific to an auditory representation, without transforming information to a motor articulatory representation. That being said, the auditory encoding that we observed in the motor cortex did not appear to be as strong as that as that observed in the STG, and exhibited comparatively weaker activity and weaker phoneme selectivity (*Figure 5—figure supplement 1b and c*, and see (*Mesgarani et al., 2014*).

In addition to having implications for perceptual models, we speculate that these results have strong implications for speech production, as auditory feedback is potentially processed directly in the vSMC in addition to the canonical auditory cortex. Speech production models currently propose a complex role for sensory feedback, where pathways exist for the activation of auditory cortex from vSMC activation (the forward prediction of production consequences), and the activation of vSMC from auditory and somatosensory input (the error correction signal) (*Guenther et al., 2006*; *Houde and Nagarajan, 2011*). In the current study, it appears that the motor cortex contains both sensory and motor representations, where the sensory representations are active during passive listening, whereas motor representations dominate during speech production.

Analysis of the time course of vSMC and STG responses revealed a heterogeneous population of both short- and longer-latencies in the inferior and superior vSMC that are generally slower than the STG (*Figure 3a–c*). Early responses in vSMC may reflect bidirectional connections from STG (*Zatorre et al., 2007*), primary auditory cortex (*Nelson et al., 2013*; *Schneider et al., 2014*) or auditory thalamus (*Henschke et al., 2014*), whereas later responses might reflect indirect connectivity in areas downstream from the STG (*Rauschecker and Scott, 2009*). Indeed, our cross-correlation analysis revealed bidirectional dynamical relationships between vSMC and STG responses, in which STG responses led vSMC responses and vice versa (*Figure 3d–f*). Still, this analysis was independent of the diverse tuning properties in the vSMC and STG electrode sets, so longer latency responses likely reflect the later responses to vowels relative to consonants. Even so, we found a wide variety of tuning and dynamical profiles in the vSMC electrodes that responded during listening. Given these proposed functional connections, activity in vSMC from speech sounds may be a consequence of sounds activating the sensory feedback circuit (*Hickok et al., 2011*). Alternatively, evoked responses in the motor cortex during passive listening may directly reflect auditory inputs arising from aggregated activity picked up by the electrode. We believe the latter scenario to be less likely, however, given that auditory responses were observed in dorsal vSMC on electrode contacts several centimeters away from auditory inputs in the STG. In addition, the spatial spread of neural signals in the high gamma range is substantially smaller than this difference – high gamma signal correlations at <2 mm spacing are only around r=0.5, and at distances of 1 cm reach a noise floor (*Chang, 2015*); Muller et al, unpublished findings). Given the observed acoustic rather than place selectivity observed during listening in the vSMC, our results suggest that motor theories of speech perception may need to be revised to incorporate a novel sensorimotor representation of sound in the vSMC.

# Materials and methods

## Participants

Nine human participants were implanted with high-density multi-electrode cortical surface arrays as part of their clinical evaluation for epilepsy surgery. The array contained 256 electrodes with 4 mm pitch. Arrays were implanted on the lateral left hemispheres over the peri-Sylvian cortex, but exact placement was determined entirely by clinical indications (*Figure 1—figure supplement 1* and *Figure 1—figure supplement 2*). Using anatomic image fusion software from BrainLab (Munich, Germany), electrode positions were extracted from the computed tomography (CT) scan, co-registered with the patient's MRI and then superimposed on the participant's 3D MRI surface reconstruction image. All participants were left hemisphere language dominant, as assessed by the Wada test. Participants had self-reported normal hearing. The study protocol was approved by the UC San Francisco Committee on Human Research, and all participants provided written informed consent.

## Task

Participants completed three separate tasks that were designed to sample a range of phonetic features. First, participants listened to eight consonant-vowel (CV) syllables (/ba/, /da/, /ga/, /pa/, /ta/, /ka/, /ʃa/, /sa/) produced by a male speaker unknown to the participant. Stimuli were presented randomly, with 4–21 repetitions of each CV syllable for 5 out of the 9 subjects included in all subsequent analyses, and one repetition of each CV syllable for 4 subjects shown only in *Figure 1—figure supplement 2*. To remain alert, participants were asked to identify the syllable they heard by selecting from a multiple-choice question on a computer with their ipsilateral (left) hand. In the second task, participants spoke aloud the same CV syllables prompted by a visual cue on the laptop computer display. In the third task, participants passively listened to natural speech samples from a phonemically transcribed continuous speech corpus (TIMIT). We chose 499 unique sentences from 400 different male and female speakers. Each sentence was repeated two times. For the phoneme selectivity analysis, we chose a subset of TIMIT phonemes that occurred more than 30 times. This resulted in an analysis of 33 phonemes. For spectrotemporal receptive field analysis (see below), data from all sentences were used.

## Data acquisition and preprocessing

Electrocorticographic (ECoG) signals were recorded with a multichannel PZ2 amplifier connected to an RZ2 digital signal acquisition system (Tucker-Davis Technologies, Alachua, FL, USA) sampling at 3,052 Hz. The produced speech was recorded with a microphone, digitized, and simultaneously recorded. The speech sound signals were presented monaurally from loudspeakers at a comfortable level, digitized, and also simultaneously recorded with the ECoG signals.

Line noise (60 Hz and harmonics at 120 and 180 Hz) was next removed from the signal with a notch filter. Each time series was visually and quantitatively inspected for excessive noise, and was subsequently removed from further analyses if its periodogram deviated more than two standard deviations away from the average periodogram of all other time series. The remaining time series were then common-average referenced (CAR) and used for analyses. The CAR was taken across 16 channel banks in order to remove non-neural electrical noise from shared inputs to the PZ2. We find that this method of CAR significantly reduces movement-related and other non-neural artifacts while not adversely affecting our signals of interest. The analytic amplitude of each time series was extracted using eight bandpass filters (Gaussian filters, logarithmically increasing center frequencies (70–150 Hz) with semi-logarithmically increasing bandwidths) with the Hilbert transform. The high-gamma power was calculated by averaging the analytic amplitude across these eight bands, and downsampling the signal to 100 Hz. The signal was finally z-scored relative to the mean and standard deviation of baseline rest data for each channel.

## Electrode selection

Supra-Sylvian cortical sites with robust evoked responses to both speech sounds and speech production were selected for this analysis. To identify if a site was responsive to speech sounds, we implemented a bootstrap t-test comparing a site's responses randomly sampled over time during speech sound presentations to responses randomly sampled over time during pre-stimulus silent intervals

(p<0.01). This resulted in 10, 22, 29, 27, and 27 sites for the five participants (n=115). Next we implemented a bootstrap t-test comparing neural responses during speech production and pre-stimulus silence (p<0.01), resulting in 25, 74, 87, 92, and 84 sites (n=362). Finally, we took the intersection of these two groups to arrive at our final supra-Sylvian sites set of 8, 16, 28, 22, and 24 sites active during listening and speaking (n=98).

To analyze the responses of the auditory cortex, we restricted the infra-Sylvian cortical sites to those that were reliably evoked by speech sounds (p<0.01, t-test between silence and speech sounds neural responses). This resulted in 73, 61, 40, 77, and 89 infra-Sylvian temporal cortical sites (n=340) responsive to speech sounds.

## Spatial clustering analysis

To investigate the degree of spatial clustering in the vSMC electrodes responsive during listening, we used the Dip-means method (*Kalogeratos and Likas, 2012*), which allows us to test whether data shows any form of clustering. Importantly, unlike the silhouette index, this allows us to distinguish between k=1 and k>1 clusters. For each subject, the pairwise distances between the spatial locations of all electrodes in a single subject were computed. Using each electrode in turn as a 'viewer' (*Kalogeratos and Likas, 2012*), we tested to see whether the distribution of distances to that electrode significantly deviated from unimodality (*Hartigan and Hartigan, 1985*). If one or more electrodes showed a signficantly non-unimodal pairwise distance histogram, then the data were considered to be clustered. Following this procedure, k-means clustering was performed with k=2 through k=6 clusters, and the silhouette index was used to determine the best number of clusters for a given subject. The silhouette index for a given data point is defined as

$$s(i) = \frac{b(i) - a(i)}{\max\{a(i), b(i)\}}$$

where b(i) is the lowest average distance of i to any other cluster of which i is not a member, and a(i) is the average distance between i and any other data point assigned to the same cluster. The silhouette index ranges from −1 to 1, with higher positive values indicating good clustering.

## Average neural response and peak high-gamma measurement

For the speaking and listening CV syllable tasks, the start of the syllable acoustics was used to align the responses of each electrode site. For the phoneme responses, the TIMIT phonetic transcriptions were used to align responses to the phoneme onset. Once responses were aligned to a stimulus, the average activity for each site to each stimulus was measured by taking the mean response over different trials of the same stimuli. The maximum of the mean responses to different stimuli were then used to measure the peak-high gamma distributions between different tasks and sites.

## Response latency analysis

We measured the onset latencies for responses to listening in STG and vSMC by calculating the average z-scored high gamma activity across all CV syllables, and then calculating the first time at which activity was significantly higher than the 500-ms pre-stimulus silent rest period (one-tailed Wilcoxon rank sum test, p<0.001). We also calculated the peak latency as the time at which the average z-scored response reached its maximum value. Differences in onset and peak latencies were compared across STG, inferior, and superior vSMC using the a two-tailed Wilcoxon rank sum test at a significance level of p<0.05 (uncorrected).

## Cross-correlation analysis

To measure the timing/dynamics between pairs of vSMC and STG sites during CV syllable listening, we performed a cross-correlation analysis between pairs of electrodes in these two regions. The cross-correlation measures the similarity of two time series at different time lags by taking pairs of electrode responses and calculating the correlation between one response and a time-shifted version of the second response. If the peak in the cross-correlation between an STG electrode and a vSMC electrode occurs at a negative lag, this indicates that the STG response leads (occurs earlier than) the vSMC response and that STG activity in the past is predictive of future activity in the vSMC. In contrast, if the peak in the cross-correlation between an STG electrode and a vSMC

electrode occurs at a positive lag, this indicates that the vSMC response leads (occurs earlier than) the STG response. The cross-correlation at time lag τ is calculated between the response at an STG electrode (denoted x) and the response at a vSMC electrode (y) as follows:

$$x * y[\tau] \overset{\text{def}}{=} \sum_{t=-0.5\,s}^{t=1\,s} x^*[t]y[t+\tau]$$

Where the maximum lag τ was chosen to be 0.75 s. Cross-correlations were normalized by $\frac{1}{M-|\tau|}$ (where M is the total number of time points in the response) to obtain an unbiased estimate at each time lag τ. The cross-correlation between vSMC and STG electrodes was calculated separately for each CV syllable trial, and then averaged across trials (see examples in *Figure 3d*).

To determine the incidence of relationships within our electrode population where STG leads vSMC, vSMC leads STG, or both are coactive, we calculated an asymmetry index. This index ranges from −1 to 1 and describes the relative power in the positive versus negative lags for each vSMC electrode. It is calculated for each vSMC electrode by taking the sum of the positive cross-correlations in the negative lags and the sum of the positive cross-correlations in the positive lags, and then computing the ratio:

$$\text{asymmetry index} = \frac{P_{\text{pos}} - P_{\text{neg}}}{P_{\text{pos}} + P_{\text{neg}}}$$

For a given vSMC electrode, an asymmetry index of −1 indicates that the cross-correlations lie fully in the negative lags (indicating that STG responses lead the vSMC response in that electrode). In contrast, a value of 1 indicates that the cross-correlations are in the positive lags only, indicating that the vSMC electrode leads all STG electrodes.

## Multidimensional scaling (MDS) analysis

To examine the relational organization of the neural responses to syllables, we applied unsupervised multidimensional scaling (MDS) to the distance matrix of the mean neural responses at the sites of interest described in Materials and methods: Electrode selection. For analysis of speaking and listening responses, the vSMC sites used were those identified as significantly active during both speech production and speech perception (n=98, *Figure 4b,c,f*). However, clustering results for speaking were similar when all vSMC sites identified as significantly active during speech production were included (n=362, *Figure 4—figure supplement 3c*). The STG sites used were those identified as significantly active during speech perception (n=340, *Figure 4e*, *Figure 4—figure supplement 3b*). Syllables placed closer together in MDS space elicited similar neural response patterns, and those further apart from one another elicited more dissimilar patterns. To calculate the distance between a pair of mean neural responses, a mean neural response to one syllable was linearly correlated to another, and the resulting correlation coefficient was subtracted from 1.

## Neural clustering analysis

We used unsupervised K-means clustering to examine the grouping of the mean neural activity to syllables of the electrodes of interest described in Methods: Electrode selection. We clustered the mean activity into 3 distinct clusters. This number of clusters was chosen because there are 3 major place of articulations and manner of articulations in the syllable stimuli set (*Figure 4a*) that have been shown to play a major role in the neural organization of motor cortex during speech production and auditory cortex during speech perception.

After clustering the neural responses into three distinct groups, we measured the similarity of the grouping to the linguistically defined grouping of consonants by place of articulation and acoustic features (*Figure 4a* and *Figure 2—figure supplement 1*) using the adjusted Rand Index (RI_adj). The RI_adj is frequently used in statistics for cluster validation. It measures the amount of agreement between two clustering schemes: one by a given clustering process (e.g. K-means), and the other by some external criteria, or gold-standard (e.g. place of articulation linguistic features). The RI_adj takes an intuitive approach to measuring cluster similarity by counting the number of pairs of objects classified in the same cluster under both clustering schemes, and controlling for chance (hence, 'adjusted' RI). It has an expected value of 0 for independent clusterings, and a maximum value of 1 for identical clustering. It is defined as the following:

Let S be a set of $n$ objects, $S = (o_1, o_2, \ldots, o_n)$. Partitioning the objects in two different ways such that $U = (U_1, \ldots, U_r)$ is a partition of $S$ into $r$ subsets, and $V = (V_1, \ldots, V_t)$ is a partition of $S$ into $t$ subsets, let:

$a$ = number of pair of objects that are in the same set in $U$ and in the same set in $V$,
$b$ = number of pair of objects that are in the same set in $U$ and in different sets in $V$,
$c$ = number of pair of objects that are in different sets in $U$ and in the same set in $V$,
$d$ = number of pair of objects that are in different sets in $U$ and in different sets in $V$.

Without adjusting for chance, the RI is simply:

$$RI = \frac{a+d}{a+b+c+d} = \frac{a+d}{\binom{n}{2}}.$$

Taking into account chance pairings, $RI_{adj}$ becomes:

$$RI_{adj} = \frac{\binom{n}{2}(a+d) - [(a+b)(a+c) + (c+d)(b+d)]}{\binom{n}{2}^2 - [(a+b)(a+c) + (c+d)(b+d)]}.$$

To localize an unbiased time window for analysis, the $\Delta RI_{adj}$ metric was derived for all time windows by subtracting the $RI_{adj}$ measured with the place of articulation features gold-standard from the $RI_{adj}$ measured with the acoustic feature gold-standard (*Figure 4—figure supplement 1*). An $\Delta RI_{adj} = 1$ denotes organization by acoustic features, and an $\Delta RI_{adj} = -1$ denotes organization by place features. The significance of the $\Delta RI_{adj}$ was computed by calculating the $RI_{adj}$ for a randomized labeling of neural responses compared to either acoustic feature or place feature clustering, taking the difference ($\Delta RI_{adj}$), and repeating this procedure 1000 times with different randomized labelings to create a null distribution of $\Delta RI_{adj}$ values. The p-value was calculated as the number of times this random $\Delta RI_{adj}$ exceeded the observed $\Delta RI_{adj}$, and was thresholded at an FDR-corrected p<0.05 using the Benjamini-Hochberg procedure (*Benjamini and Hochberg, 1995*).

## Electrode phoneme selectivity index (PSI)

To characterize the phoneme selectivity of each electrode site, we implemented the PSI calculation described by *Mesgarani et al., 2014*. In short, for a single site, we summed the number of responses that were statistically different (Wilcoxon rank-sum test, p<0.01, corrected for multiple comparisons) from the response to a particular phoneme. This resulted in a PSI that ranges from 0 to 32, where a PSI = 32 is an extremely selective electrode and a PSI = 0 is not selective. A PSI describes an electrode's selectivity to one phoneme, and a vector of PSIs describes an electrode's selectivity profile to all phonemes.

## Spectrotemporal receptive field (STRF) estimation

The spectrotemporal representation of speech sounds was first estimated using a cochlear frequency model, consisting of a bank of logarithmically spaced constant Q asymmetric filters. The filter bank output was subjected to nonlinear compression, followed by a first order derivative along the spectral axis modeling a lateral inhibitory network, and an envelope estimation operation (*Wang and Shamma, 1994*). This resulted in a two dimensional spectrotemporal representation (spectrogram) of speech sounds simulating the pattern of activity on the auditory nerve.

We then estimated the spectrotemporal receptive fields (STRFs) of the sites from passive listening to TIMIT using normalized reverse correlation (*Aertsen and Johannesma, 1981*; *Klein et al., 2000*; *Theunissen et al., 2001*; *Woolley et al., 2006*) between spectrotemporal representation of the sentences and the evoked neural activity (STRFLab software package: http://strflab.berkeley.edu, *DirectFit* routine). The STRF is a linear filter that describes which combinations of spectrotemporal features will elicit a neural response in a given electrode. The relationship between the STRF, $H$, stimulus spectrogram, $S$ (as estimated above), and the predicted response, $\hat{r}(t)$, of an electrode are given by the following equation:

$$\hat{r}(t) = \sum_{i=0}^{M-1} \sum_{\tau=0}^{N-1} H(\tau,f)S(t - \tau,f)$$

where N is the number of delays of length τ after which the STRF will be estimated (reflecting memory for the stimulus), and M is the number of frequency bands in the spectrogram. To estimate the STRF, we minimize the mean squared error between the predicted and observed responses. To prevent overfitting, we used an L2 regularization procedure in which a ridge hyperparameter and sparseness hyperparameter were calculated for each electrode's STRF (details in [*Woolley et al., 2006*]). The ridge hyperparameter acts as a smoothing factor on the STRF, whereas the sparseness hyperparameter controls the number of non-zero weights in the STRF. These hyperparameters were optimized with a systematic hyperparameter grid search maximizing for mutual information (bits/s). With the optimized hyperparameters, we calculated the final STRF and correlation between the predicted and actual neural response using cross-validation. To do this, a STRF was derived using 9/10 of the stimuli-response pairs, and the Pearson correlation coefficient (indicating the STRF goodness-of-fit) was measured by predicting the remaining one-tenth responses. This was repeated 10 times with 10 non-overlapping stimuli-response pair sets. The final STRF and correlation number were derived by averaging the 10 STRFs and correlation coefficients.

## Note on statistical tests

To assess statistical differences, we used independent sample t-tests when the data were found not to deviate significantly from normality (KS test). When data were not normally distributed, we used the nonparametric Wilcoxon rank sum test. In some cases, a bootstrap t-test was used.

## Acknowledgements

N Mesgarani, M Leonard, and J Houde provided helpful comments on the manuscript. EFC was funded by the US National Institutes of Health grants R01-DC012379, R00-NS065120, DP2-OD00862, and the Ester A and Joseph Klingenstein Foundation. LSH was funded by a Ruth L Kirschstein National Research Service Award (1F32DC014192-01) through the NIH National Institute on Deafness and Other Communication Disorders.

## Additional information

### Funding

| Funder | Grant reference number | Author |
|--------|------------------------|--------|
| National Institute on Deafness and Other Communication Disorders | 1F32DC014192-01 | Liberty S Hamilton |
| NIH Office of the Director | OD00862 | Edward F Chang |
| McKnight Foundation | | Edward F Chang |
| National Institute on Deafness and Other Communication Disorders | R01-DC012379 | Edward F Chang |
| National Institute of Neurological Disorders and Stroke | R00-NS065120 | Edward F Chang |

The funders had no role in study design, data collection and interpretation, or the decision to submit the work for publication.

### Author contributions

CC, Acquisition of data, Analysis and interpretation of data, Drafting or revising the article; LSH, KJ, Analysis and interpretation of data, Drafting or revising the article; EFC, Conception and design, Acquisition of data, Analysis and interpretation of data, Drafting or revising the article

## Author ORCIDs

Liberty S Hamilton, http://orcid.org/0000-0003-0182-2500
Edward F Chang, http://orcid.org/0000-0003-2480-4700

## Ethics

Human subjects: Written informed consent was obtained from all study participants. The study protocol was approved by the UCSF Committee on Human Research.

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
