## [Decision Letter]

Thank you for submitting your work entitled "The Auditory Representation of Speech Sounds in Human Motor Cortex" for consideration by *eLife*. Your article has been reviewed by two peer reviewers, including Nicholas Hatsopoulos, and the evaluation has been overseen by Barbara Shinn-Cunningham as the Reviewing Editor and David Van Essen as the Senior Editor.

The reviewers have discussed the reviews with one another and the Reviewing Editor has drafted this decision to help you prepare a revised submission.

Summary:

This study examines evoked activity in speech motor cortex due to passive listening of consonant-vowel (CV) syllables to determine the nature of these representations. High-density ECoG grids recorded local field potentials from the surface of motor and auditory cortices of epileptic patients who either produced CVs or passively listened to them. During speaking, evoked neural activity in ventral sensorimotor cortex (vSMC) was somatotopically organized according to place of articulation whereas during listening, the organized seemed to be based on acoustic features. Both reviewers and the reviewing editor believe that the study could shed light on the important question of whether or not speech perception depends upon encoding spoken language in a motor-based neural code. The innovative experimental and analytical techniques add to our enthusiasm for the work. While the data suggest that language perception is encoded in acoustic features rather than speech-motor production features, all of us felt that there were some major issues, especially with the data analysis, that need to be addressed before the full impact of the study can be evaluated.

Essential revisions:

1) The methods state that the data were analyzed using a common-average reference. Following this, to estimate the high-γ power relative to baseline, the power (extracted using the Hilbert transform) in 8 bandpass frequency regions was computed and averaged, and then turned into a z score. Given this processing stream, the number and locations of the electrodes across which the common-average reference was computed might bias the findings, especially since oscillation power was ultimately analyzed (i.e., very strong high-γ power that was synchronous across a small number of nearby sites would end up looking like weak high-γ power across all sites). This issue needs to be discussed; perhaps some bootstrap analysis or other control could demonstrate that such artifacts are not an issue.

2) The authors argue that during listening "the tuning properties of responsive sites in vSMC.… appear to give rise to an acoustic sensory organization of speech sounds (rather than purely motor organization) in motor cortex during listening." The impact of the work rests on this claim, that the responses in vSMC during listening are organized by acoustic rather than motor properties of sound. However, proving this claim requires showing that statistically, the listening response in vSMC is quantitatively better described as "auditory" rather than "motor" (i.e., by a direct statistical comparison of results plotted in Figure 3). Without this proof, the impact of this paper may be less than claimed. This should be addressed by a statistical comparison of the two fits of the vSMC_listening_ results.

3) In the analyses that are provided, the authors select different electrodes when evaluating vSMC_listening_ (98 electrodes) and vSMC_speaking_ (362 electrodes). The electrodes evaluated in the vSMC_speaking_ condition are not all active during listening. To allow direct comparison of results for speaking vs. listening, it makes more sense to select identical subsets for both analyses. This in turn means limiting the analysis to those sites that are active for both vSMC_speaking_ and vSMC_listening_ conditions. It seems problematic to claim that activity patterns are different across conditions when those patterns are being examined in different sets of electrodes. This can best be addressed by a reanalysis that directly compares vSMC_speaking_ and vSMC_listening_ responses in the more limited set of electrodes that respond in both cases.

4) Related to item 3, the MDS plots for vSMC_listening_ (Figure 3) are less clearly clustered than vSMC_speaking_. This may be an artifact of the 3x difference in sample sizes. Depending on how the authors decide to deal with item 2, this may not remain a problem; however, if there is a comparison of clustering results using different sample sizes, the authors need to verify that the results shown in Figure 3 are not an artifact of this difference (e.g., by showing that the same effect holds if the vSMC_speaking_ and STG_listening_ datasets are subsampled to equalize the sample size).

5) It may be that the evoked responses in motor cortex during passive listening reflect auditory inputs. While high-γ LFPs strongly correlate with multi-unit spiking (as the authors state), high-γ LFPs may also reflect the aggregate synaptic potentials near the electrode. The authors should discuss this possibility.

6) A motor representation in vSMC during listening may be present in a subset of the vSMC sites. The authors show that response in some vSMC sites lead STG sites (Figure 4). If the MDS analysis was restricted to sites that lead STG (i.e., vSMC signals with very short latencies), would these sites reveal a motor representation? Some vSMC may be generating motor-like responses that are triggered by the auditory stimulus, but that reflect motor, not sensory properties. Analysis of the short-latency vSMC responses would address this concern.

7) The authors report that 16 total sites in vSMC have activity that was "strongly" predicted by a linear STRF, with "strong" defined as r>0.10. The authors need to make clear where the subset of 16 electrodes with significant STRFs were located, and also whether or not this fraction of locations (16/X) is significantly above chance (e.g., correcting for false discovery rate). Most likely, X should include all auditory-responsive vSMC sites, but this is not stated clearly. In addition, the authors should explain by what criterion r>0.10 (i.e., a correlation in which 1% of the variance is explained by the strf) is a good criterion for a "strong" prediction. For instance, how does this correlation strength compare to STRF predictions in STG?

---

## [Author Response]

*Essential revisions:*

*1) The methods state that the data were analyzed using a common-average reference. Following this, to estimate the high-*γ *power relative to baseline, the power (extracted using the Hilbert transform) in 8 bandpass frequency regions was computed and averaged, and then turned into a z score. Given this processing stream, the number and locations of the electrodes across which the common-average reference was computed might bias the findings, especially since oscillation power was ultimately analyzed (i.e., very strong high-*γ *power that was synchronous across a small number of nearby sites would end up looking like weak high-*γ *power across all sites). This issue needs to be discussed; perhaps some bootstrap analysis or other control could demonstrate that such artifacts are not an issue.*

We agree that it is important to show that our use of the common average reference (CAR) does not adversely affect our results, and we have included more information in the manuscript to clarify what was done. We do not find that the CAR reduces the high-γ power, but rather serves to remove large artifacts across wide sensory and non-sensory areas that are likely non-neural in origin. In our study, the CAR was applied across 16 channel blocks running in columns across the ECoG grid before segmenting data into speech responsive or non-responsive. Blocks of 16 channels were used because when the ECoG grids are plugged into the amplifier, the 256 channels are separated into 16 channel sets, each of which is plugged into a separate bank (http://www.tdt.com/files/specs/PZ2.pdf). The grids are designed such that the 16 channel sets correspond to columns on the grid – for example, channels 1–16 share a common plug, 17-32 share the next plug, 33-48 the next, etc. Thus, when we subtract the CAR in these 16-channel banks, it removes noise that is shared across a group of wires plugged into the same bank on the amplifier. In practice, we find that this significantly reduces the amount of 60 Hz noise and removes some artifacts due to movement, but does not adversely affect signals in the high γ band. Because of the orientation of the grids, the columns of the grid generally run ventrodorsally and include electrodes over speech-selective and non-selective areas (STG and parietal cortex, or STG and motor cortex, for example). Thus, subtracting signals shared across this wide region do not generally affect the high-γ band power in this way (that is, we do not find that strong high-γ power is weakened by the CAR). As a quantitative comparison, we compared the phoneme aligned high γ matrices with and without the CAR (shown for one subject in Figure 6). There is no significant difference between the high γ data with and without the CAR (Wilcoxon sign rank test, p=1).

Author response image 1.**DOI:**
http://dx.doi.org/10.7554/eLife.12577.015

However, because of its ability to remove artifacts due to non-neural electrical noise (see Figure 7 power spectrum showing power in different neural frequency bands with and without the CAR—error indicates standard deviation across channels), we chose to leave the results in the paper as is, using the CAR as described. We added the following sentences to the Methods section to clarify the use of the CAR: "The CAR was taken across 16 channel banks in order to remove non-neural electrical noise from shared inputs to the PZ2. We find that this method of CAR significantly reduces movement-related and other non-neural artifacts while not adversely affecting our signals of interest."

Author response image 2.**DOI:**
http://dx.doi.org/10.7554/eLife.12577.016

*2) The authors argue that during listening "the tuning properties of responsive sites in vSMC*.… *appear to give rise to an acoustic sensory organization of speech sounds (rather than purely motor organization) in motor cortex during listening." The impact of the work rests on this claim, that the responses in vSMC during listening are organized by acoustic rather than motor properties of sound. However, proving this claim requires showing that statistically, the listening response in vSMC is quantitatively better described as "auditory" rather than "motor" (i.e., by a direct statistical comparison of results plotted in Figure 3). Without this proof, the impact of this paper may be less than claimed. This should be addressed by a statistical comparison of the two fits of the vSMC_listening_ results.*

We agree that this direct statistical comparison is important for interpretation of our results and thank the reviewers for the suggestion. To this end, we have performed the statistical comparison of the data in Figure 3 and 3F (now Figure 4) for vSMC electrodes during listening and found a significantly higher RI_adj_ for acoustic (manner) features compared to place of articulation features (p<0.001, Wilcoxon rank sum test). We now show quantitatively that the listening responses in vSMC are indeed better characterized as "auditory" rather than "motor". This comparison is shown in Figure 4 and described in the figure legend. In addition, we have added the following text to the Results section: "Importantly, however, clustering by acoustic manner features was significantly better than clustering by place features in vSMC electrodes during listening (p<0.001, Wilcoxon rank-sum, Figure 4)."

*3) In the analyses that are provided, the authors select different electrodes when evaluating vSMC_listening_ (98 electrodes) and vSMC_speaking_ (362 electrodes). The electrodes evaluated in the vSMC_speaking_ condition are not all active during listening. To allow direct comparison of results for speaking vs. listening, it makes more sense to select identical subsets for both analyses. This in turn means limiting the analysis to those sites that are active for both vSMC_speaking_ and vSMC_listening_ conditions. It seems problematic to claim that activity patterns are different across conditions when those patterns are being examined in different sets of electrodes. This can best be addressed by a reanalysis that directly compares vSMC_speaking_ and vSMC_listening_ responses in the more limited set of electrodes that respond in both cases.*

We agree with the reviewers that our claim that areas of vSMC show an acoustic representation while listening and a place of articulation representation while speaking would be stronger by analyzing the same subset of electrodes for both speaking and listening. We repeated the clustering analyses in Figure 4 using only the 98 electrode subset during speaking and listening, and found that our results still hold in this more limited set. Thus, we opted to replace panel b in Figure 4 with the results from the 98 electrode subset, and placed the former Figure 4 in Figure 4—figure supplement 3; Figure 4—figure supplement 2, panel c. When only the 98 electrode subset is analyzed, responses still appear to be similar for similar place of articulation – for example, /b/ and /p/, the bilabials, are close to one another in MDS space, as are the velars /k/ and /g/, and the alveolars /∫/, /s/, /t/, and /d/.

The clustering trajectory analysis in Figure 4—figure supplement 1 has also been replaced using the same electrodes for speaking and listening. Responses are aligned according to the acoustic onset of the CV syllable (at t=0 s). Significant ∆RI_adj_ values are indicated by the blue and red windows (FDR-corrected p<0.05, permutation test). Here it is clear that, just prior to and during speaking, vSMC responses cluster according to place of articulation. During listening, however, those same responses exhibit selectivity according to manner of articulation, reflecting an acoustic representation.

*4) Related to item 3, the MDS plots for vSMC_listening_ (Figure 3) are less clearly clustered than vSMC_speaking_. This may be an artifact of the 3x difference in sample sizes. Depending on how the authors decide to deal with item 2, this may not remain a problem; however, if there is a comparison of clustering results using different sample sizes, the authors need to verify that the results shown in Figure 3 are not an artifact of this difference (e.g., by showing that the same effect holds if the vSMC_speaking_ and STG_listening_ datasets are subsampled to equalize the sample size).*

We agree with the reviewers that the strong clustering observed in the vSMC speaking data may be due to the larger sample size, however, in reanalyzing the speaking data as described in the response to point (3), we found that clustering according to place of articulation was still observed when the speaking data was limited to the same set of electrodes used in the vSMC listening analysis. Thus, we have chosen to present the clustering data using the same sample size for vSMC listening and speaking, and present for reference the original analysis with speaking data for all active electrodes in Figure 4—figure supplement 2.

*5) It may be that the evoked responses in motor cortex during passive listening reflect auditory inputs. While high-*γ *LFPs strongly correlate with multi-unit spiking (as the authors state), high-*γ *LFPs may also reflect the aggregate synaptic potentials near the electrode. The authors should discuss this possibility.*

We thank the reviewers for the suggestion, and have added to the last paragraph of the Discussion section additional details on where such auditory signals may arise. While it is possible that some inferior vSMC electrodes are picking up on auditory inputs, we believe it to be unlikely based on other analyses we have performed in the lab on spatial correlations across the ECoG grid at different neural frequency bands. We have added the following text:

"Alternatively, evoked responses in the motor cortex during passive listening may directly reflect auditory inputs arising from aggregated activity picked up by the electrode. We believe the latter scenario to be less likely, however, given that auditory responses were observed in dorsal vSMC on electrode contacts several centimeters away from auditory inputs in the STG. In addition, the spatial spread of neural signals in the high γ range is substantially smaller than this difference – high γ signal correlations at <2 mm spacing are only around r=0.5, and at distances of 1 cm reach a noise floor (Chang EF, Neuron 2015; Muller et al., in revision). Given the observed acoustic rather than place selectivity observed during listening in the vSMC, our results suggest that motor theories of speech perception may need to be revised to incorporate a novel sensorimotor representation of sound in the vSMC."

*6) A motor representation in vSMC during listening may be present in a subset of the vSMC sites. The authors show that response in some vSMC sites lead STG sites (Figure 4). If the MDS analysis was restricted to sites that lead STG (i.e., vSMC signals with very short latencies), would these sites reveal a motor representation? Some vSMC may be generating motor-like responses that are triggered by the auditory stimulus, but that reflect motor, not sensory properties. Analysis of the short-latency vSMC responses would address this concern.*

We agree that this is an interesting additional analysis to determine whether the short-latency sites in the vSMC show differences in their motor vs. sensory responses. We performed the MDS analysis as suggested by restricting the analysis to sites that lead the STG as evidenced by a positive asymmetry index (see Figure 3 and Methods section– Cross Correlation analysis). This analysis yielded 50 vSMC electrodes. We found that even in this subset of short-latency responders, responses during listening appeared to cluster according to acoustic rather than articulatory features. This suggests that vSMC does not generate motor-like responses during listening. These results are now shown in Figure 4—figure supplement 3.

*7) The authors report that 16 total sites in vSMC have activity that was "strongly" predicted by a linear STRF, with "strong" defined as r>0.10. The authors need to make clear where the subset of 16 electrodes with significant STRFs were located, and also whether or not this fraction of locations (16/X) is significantly above chance (e.g., correcting for false discovery rate). Most likely, X should include all auditory-responsive vSMC sites, but this is not stated clearly. In addition, the authors should explain by what criterion r>0.10 (i.e., a correlation in which 1% of the variance is explained by the strf) is a good criterion for a "strong" prediction. For instance, how does this correlation strength compare to STRF predictions in STG?*

We appreciate the reviewers’ comments and have attempted to clarify the results of the STRF analysis in the new manuscript, as well as address the issue of correction for multiple comparisons. We recalculated our STRF correlations using a shuffled permutation test in which we take the predicted response from the STRF and shuffle it in time, using 500 ms chunks to maintain some of the temporal correlations inherent in the ECoG signal. We then calculate the correlation between the shuffled predictions and the actual response to get a sense of the magnitude of STRF correlations that would be observed due to chance. We perform this shuffling procedure 1000 times and calculate the p-value as the percentage of times that the shuffled time series correlations are higher than the correlations from our STRF prediction. We then perform FDR correction using the Benjamini-Hochberg procedure. As it turns out, there are many STRF correlations with r-values less than 0.1 that are still significantly higher than what would be expected due to chance. Still, we believe r = 0.1 to be a reasonable cutoff and have opted not to include more sites with lower correlation values. We have toned down the claim that these sites are "strongly" predicted by a linear STRF, instead writing “A fraction of vSMC sites (16/98 sites total) were reasonably well-predicted with a linear STRF (r>=0.10, p<0.01) (Theunissen et al., 2001).” Although this fraction is relatively small, these sites may encode auditory-related features that are not spectrotemporal in nature, or may respond nonlinearly to auditory features, and thus would not be captured appropriately by the STRF. In addition, more typical auditory areas including the anterior STG can show significant auditory responses that are not adequately captured by a linear STRF. The electrodes described here were located in both superior and inferior vSMC, as shown in Figure 5, which includes STRFs from all 5 subjects plotted as a function of the rostral-caudal distance from the central sulcus and the dorsal distance from the Sylvian fissure in millimeters.

On average, the correlation strength in the vSMC electrodes is lower than the correlation strength in STG electrodes, as shown by the new panel b added to Figure 5 and additional text in the Results section: “Still, the prediction performance of STRFs in vSMC was generally lower than that of the STG (Figure 5).”